# Energy Intake Evaluation by a Learning Approach Using the Number of Food Portions and Body Weight

**DOI:** 10.3390/foods10102273

**Published:** 2021-09-26

**Authors:** Sylvie Rousset, Sébastien Médard, Gérard Fleury, Anthony Fardet, Olivier Goutet, Philippe Lacomme

**Affiliations:** 1Unité de Nutrition Humaine, University Clermont Auvergne, UNH, UMR1019, INRAE, 63000 Clermont-Ferrand, France; anthony.fardet@inrae.fr; 2Laboratoire d’Informatique (LIMOS, UMR CNRS 6158), University Clermont Auvergne, 63000 Clermont-Ferrand, France; sebastien.medard@etu.uca.fr (S.M.); gyfleury@gmail.com (G.F.); placomme@isima.fr (P.L.); 3Openium, 15 rue Jean Claret Bâtiment le XV, La Pardieu, 63000 Clermont-Ferrand, France; o.goutet@openium.fr

**Keywords:** prediction of energy intake, total number of food portions, body mass index, energy expenditure, dietary apps

## Abstract

An accurate quantification of energy intake is critical; however, under-reporting is frequent. The aim of this study was to develop an indirect statistical method of the total energy intake estimation based on gender, weight, and the number of portions. The energy intake prediction was developed and evaluated for validity using energy expenditure. Subjects with various BMIs were recruited and assigned either in the training or the test group. The mean energy provided by a portion was evaluated by linear regression models from the training group. The absolute values of the error between the energy intake estimation and the energy expenditure measurement were calculated for each subject, by subgroup and for the whole group. The performance of the models was determined using the test dataset. As the number of portions is the only variable used in the model, the error was 26.5%. After adding body weight in the model, the error decreased to 8.8% and 10.8% for the normal-weight women and men, respectively, and 11.7% and 12.8% for the overweight women and men, respectively. The results prove that a statistical approach and knowledge of the usual number of portions and body weight is effective and sufficient to obtain a precise evaluation of energy intake after a simple and brief enquiry.

## 1. Introduction

The evaluation of energy intake is commonly performed using the 24-h dietary recall or frequency questionnaire, or 3- to 7-day reported food intake [1,2]. Doubly labelled water (DLW) is used as a reference method to measure total energy intake (TEE) in free-living conditions and to validate reported energy intake in many studies, including [3,4]. This reference methodology is based on the fundamental principle of energy balance, meaning that total energy expenditure (TEE) is equal to energy intake when the body weight is stable (in the absence of a significant weight change) [5]. Many authors found a positive correlation between TEE measured by DLW and body weight, but a flat slope between TEE and reported energy intake [6,7,8]. According to these authors, the underestimation of energy intake concurrent with increasing weight may be due to the imitation error of the food reported by the general population. That means that food intake is reported in the same way, regardless of the body weight range. This is confirmed by Novotny et al. (2003), who found an overall underreporting of 294 kcal/d energy intake [9]. This underestimation of energy intake was higher in women than in men: 85% of women underreported their food intake by 621 kcal/d, whereas 61% of men underreported theirs by 581 kcal/d. In contrast, 15% of women overreported their energy intake by 304 kcal/d and 39% of men by 683 kcal/d. The poor food intake estimation was mainly related to body fat mass and body dissatisfaction. The higher the body fat percentage was, the higher the underreporting of energy intake was [9]. Gender also played an important part in the correct estimation of food intake, with men being better estimators than women [9]. Many other studies have proved that both underreporting and overreporting occur, regardless of the methods used for food intake assessments [2,10].

Since the cost of the DLW method is a liming factor for large-scale studies, such as epidemiological ones, there is a great interest to replace DLW with another less costly technique or procedure able to estimate energy intake with a high level of accuracy. The development of the new information and communication technologies and the widespread use of smartphones open new application prospects in nutrition and dietary assessment. For example, the use of dietary mobile applications led to a decrease in weight, waist circumference, and energy intake compared to control in adults with chronic diseases [11]. Researchers also believe that technology can improve diary reporting by reducing memory and representation bias and errors from data processing [12]. In several studies, subjects were told to take photographs with their smartphones in order to improve reporting and avoid food omission. However, there were many problems with the quality, the angle, and the lack of pictures or descriptive comments associated with the picture [13]. Pendergast et al. (2017) used the smartphone meal diary app (FoodNow) to measure food intake and compare the energy intake estimation with the total energy expenditure provided by an accurate physical activity research monitor (SenseWear Armband) in a population of young people with a healthy BMI range [14]. The authors demonstrated that there is a high correlation coefficient between the estimated energy intake and the measured energy expenditure. The mean difference between the estimation and the measurement was 197 kcal/d for a mean energy expenditure of 2395 kcal/d, i.e., an underestimation of energy intake by about 8%. However, they showed wide levels of agreement between the two methods (Armband and FoodNow app) at the individual level (−886 kcal to +491 kcal). The authors concluded that the app is a more suitable tool for estimating the mean energy intake of a group rather than that of an individual. A recent article on food evaluation through smartphones showed that smartphone applications provided similar but not better validity or reliability when their results were compared with classical dietary assessments [14].

Further work is necessary to improve information and communication technology (ICT) tools used for in-depth evaluation. The improvement of food intake evaluation should focus on data collection as well as data treatment. In this study, we propose a simple model of energy intake estimation with a satisfactory level of accuracy. 

## 2. Materials and Methods

### 2.1. Subjects

This observational study was conducted on 190 subjects who were recruited anonymously for the open-door event of an INRAE center and through social networks. The subjects were required to adults (older than 18 years), have an Android smartphone, and provide consent to participate in the study, which lasted four days. Moreover, we asked them to fill in their personal and dietary information honestly in the App.

A total of 116 women and 74 men, either normal weight (NW, n = 123), or overweight (OW, n = 67), were studied in free-living conditions (Table 1). A total of 131 were used for model development (35 normal weight men, 52 normal weight women, 18 overweight men and 26 overweight women, training group) and 59 subjects (12 normal weight men, 25 normal weight women, 9 overweight men and 13 overweight women, test group) were used to evaluate the validity of the energy intake estimation. 

### 2.2. Data Collection and Energy Balance Principle

The subjects downloaded the WellBeNet app at the Play Store and informed the researcher about their age and gender. They logged their height and weight in the app. They were then asked to use the eMouve and NutriQuantic parts of the WellBeNet application for four consecutive days (three weekdays and one day during the weekend). They were told to wear the smartphone in their pant pocket to collect accelerometry data for the waking period. eMouve provides an accurate estimation of total energy expenditure (TEE) in normal-weight and overweight subjects, i.e., 5.7% and 8.5% of error in absolute value [15,16].

NutriQuantic was used to collect the number of portions consumed, regardless of the food category, during the same period. A guide for the estimation of a portion was sent to each subject. For each food category, examples of portion size were given: one portion of fruit is provided by a large fruit (100–150 g) or two small fruits (50–75 g); one portion of raw or cooked vegetables fills of 1/3 of a plate (150–250 g); one portion of starchy food is represented by a slice of bread or two slices of toast or of wheat loaf; one portion of dairy product is provided by a yogurt (125 g) or white cheese (100 g), or a glass of milk (100–125 g), or a piece of cheese (30–50 g), etc.

A nutritional score was assigned to each of the 11 food categories based on the number of portions and according to French and international nutritional guidelines [17]. The score varies between 0 and 1. The nutritional balance score of the diet is the result of a confidential calculation over the 11 food categories [18].

Energy balance is based on the fundamental principle that energy intake (EI) is equal to energy expenditure during a stable body weight period [5]. 

### 2.3. Ethical Approval

This observational study was conducted according to the guidelines laid down in the Declaration of Helsinki and the French legislation for the collection of anonymous human data. Written or verbal informed consent was obtained from all subjects for the aggregated treatment of their data. Verbal consent was witnessed and formally recorded. No specific ethical approval was necessary for this study.

### 2.4. Statistical Models

A Chi-2 test (χ^2^) was used to compare the distribution of men/women, and normal-weight/overweight individuals in the two populations (training and test groups). Statistical significance was set at *p* < 0.05.

For each gender, a one-way analysis of variance model (GLM) was carried out to determine the effects of BMI status (normal weight vs. overweight) on age, height, weight, number of portions per day, nutritional balance score, and daily TEE. The GLM (general linear model) procedure is appropriate for treating unbalanced data. We used the Type III sum of squares that measures the differences between predicted weight status means over a balanced weight status population. A mean comparison test (LSMeans) was carried out when *p* < 0.05. SAS software, version 9.4, was used to carry out the frequency test and analysis of variance.

In the first step, a linear regression model was developed from the data collected by all the subjects of the training group. The model used only the total mean number of portions per day, to explain the energy expenditure. All data collected by the subjects in the test group were used to validate this first prediction model. In the second step, four models (2×2) were developed by gender and BMI subgroups from the training group data and validated on the test group data. The subjects in the test group were different from those in the training group. The regression models were implemented in Python to compute the model errors in absolute value in both training and test groups and to assess the energy intake of each subject. Two constraints were added to the solutions given by the regression models: their values have to be positive (for the number of portions and weight) or null (weight). The value of the coefficient for the number of portions has to be positive because each food portion provides energy. The constraint on body weight is assumed to be lower: if the number of portions could completely account for the energy intake, then weight could have a negligible effect on energy intake. In this case, the value of the coefficient will take the null value, if not a positive value. The values of coefficients were determined from the data collected by the training group: normal-weight and overweight men and normal-weight and overweight women. The values of these coefficients were then applied to the data of the test group for validation.

Agreement between the energy intake (EI) and TEE was evaluated by Bland–Altman plots [19]. The plots were drawn up showing the mean difference between estimated EI values and TEE values provided by eMouve against the mean of the two methods. The bias is estimated by the mean difference (M) and the standard deviation (s). Statistically, 95% of the differences will range between M ± 2 s (agreement limits). The validity of EI was evaluated in each regression model by comparing the agreement level between the EI and TEE.

## 3. Results

### 3.1. Differences between BMI Statuses

The training and test groups were similar in gender and BMI status distribution (χ^2^ = 0.40, *p* = 0.52, χ^2^ = 0.69, *p* = 0.81, respectively). There was no difference in the BMI status distribution between men and women (χ^2^ = 0.35, *p* = 0.55, Table 1).

The one-way analysis of variance showed that age, body weight, nutritional balance score, and total energy expenditure differ between normal-weight and overweight women (Table 2). Overweight women were older (43 years vs. 37 years) and their body weight (85 kg vs. 57 kg, *p* < 0.0001) and energy expenditure were higher (Table 2). They took a number of food portions similar to that of normal-weight women, but their nutritional score was lower than that of normal-weight women (Table 2). There was no significant difference in age for men (38 years vs. 34 years), or in the number of portions between the two weight statuses (11.2 portions/d vs. 12.7 portions/d). The significant differences observed were that body weight (94 kg vs. 69 kg, *p* < 0.0001) and energy expenditure were higher in the overweight subgroups (Table 2). 

### 3.2. Errors of Regression Models and Agreement with Total Energy Expenditure 

The first model included only one variable: the total number of portions/d. 

EI1i =174.8×Pi + Ei: First model for all the subjects.

Pi: Number of portions (mean number/d), and Ei_:_ error for an individual i.

The estimated energy intake of a portion was 174.8 kcal. The error in absolute value was 30.7% and 26.5% in the training and test populations, respectively. The Bland–Altman plots show that all the points except six (in the training group) and two (in the test group) are included between the lower and upper limits of agreement (Mean + 2 SD, Mean − 2 SD, Figure 1). 

The bias, equal to −287 and −324 kcal/d in the two subject groups, indicated that the EI_1_ was underestimated by about 10%, but the 95% limits of agreement were wide (−2199 kcal/d to 1611 kcal/d and −1825 to 1176 kcal/d). The first model did not provide satisfactory results on individual energy intakes because of the large gaps between estimated energy intake and energy expenditure. 

Since the energy intake of women and normal-weight subgroups is lower than that of men and overweight subgroups, and because underreporting is frequent as body mass index increases, we performed four status regression models (for each gender and weight status) with two explanatory energy intake variables: number of portions and body weight. 

The values of regression coefficients for the number of portions and the body weight determined in the training group are shown in Table 3. All the coefficients for weight are positive. The energy intake of normal-weight (70 kg) and overweight men (93 kg) is explained by weight contribution (33.7×70  = 2359 and 27.1×93=2520) and by the food portions (22.1×13.1=289 kcal and 36.7×11.2=411 kcal). The energy intake of normal-weight and overweight women (58 kg and 83 kg on average) is explained by weight (1757 kcal and 1978 kcal) and food portions (261 kcal and 352 kcal). These results show that body weight explained 90% and 87% of energy intake in normal-weight men and women, and 86% and 85% in overweight men and women, respectively. These results also showed that the higher energy intake observed in overweight compared to normal-weight subjects can be explained by a higher number of food portions and body weight contributions. Thus, the energy intake estimated from both food portions and body weight increased in both overweight men and women (Table 3). 

EI2i =33.7×Wi +22.1×Pi+ Ei: Second model for normal-weight men

EI3i =27.1×Wi +36.7×Pi + Ei: Third model for overweight men

EI4i =30.3×Wi +22.3×Pi +Ei: Fourth model for normal-weight women

EI5i =23.8×Wi +32.3×Pi + Ei: Fifth model for overweight women

where Wi: weight (kg), Pi: number of portions (mean number/d), and Ei: error for an individual i.

For normal-weight men, the errors in absolute value fell to 11.5% and 8.2%, respectively, in the training and test groups. The bias is close to zero (78 and 9 kcal/d, Figure 2) with 95% agreement limits, which is half the size of the full group bias. Only two individuals are located outside the agreement limits in the training group and none in the test group. The two outliers reported 30 portions/d and 5.5 portions/d. The gap between energy intake (EI2) and TEE is lower than 400 kcal/d in most of the subjects in the training group and in all of the subjects in the test group (Figure 2). 

The errors in absolute value are 11.5% and 7.6%, respectively, in the training and test groups with overweight men. The bias is close to zero (55 and −20 kcal/d, Figure 3). The 95% agreement limits are slightly higher than for the normal-weight men but only one individual is located outside the agreement limits in the training group and none in the test group. As for the normal-weight men, the (EI3 − TEE) gap is frequently lower than 400 kcal/d. 

The gaps between EI4 and TEE are 10.3% and 8.1% in absolute value for the normal-weight women. The bias is close to zero (29 kcal/d and −95 kcal/d for the training and test groups, respectively, Figure 4). Only three subjects in the training group are outside the agreement limits. One of them reported a very low number of portions: 4.75 portions/d. All the women belonging to the test group had an energy intake close to TEE (±300 kcal/d).

For the overweight women, the difference between estimated EI5 and TEE was 9.1% and 5.8% in the training and test groups. The bias is close to zero (−17 and 91 kcal/d, Figure 5). None of the subjects are outside the agreement limits and only one is in the test group. This woman reported a higher number of portions than the average (17.7 portions/d vs. 11.1 portions/d). Most of the overweight women had an estimated intake equal to TEE ± 300 kcal/d. 

## 4. Discussion

The aim of this study was to assess energy intake on the basis of simple variables. Our work proves that two variables are highly significant to reach this objective: body weight and the reported number of food portions. This study proves that the number of portions was not significantly different between gender or BMI status; the mean number of standardized portions reported for the general population was between 10 and 13 portions/d [6]. Since energy requirements are known to be higher in men than in women, and higher in overweight than in normal-weight people, the size and/or the energy content of the portion could differ between them [20]. Models of regression were performed, taking account of the number of portions in the 11 food categories (results not shown), but the estimations of energy intake were not better than those of the total number of food portions. The findings of Kelly et al. (2008) may explain our results: they observed that an increased risk of obesity may not be associated with specific foods/food groups but rather with an overall increase in the range of foods and food groups being consumed [21].

The difference in portion size is probably an explanatory factor for the lack of association between the number of portions, portion size, and BMI status. Even though we gave the subjects a guide to evaluate the portion size, the subjects used their own references to determine the portion unit. Ledikwe et al. (2005) and Bhupathiraju and Hue (2016) found that a large food portion size was associated with obesity in America [22,23]. The overall energy intake increased by 35% when food portion size doubled [22]. In contrast, a regular food portion size contributes to adequate energy intake and, consequently, weight maintenance. 

Another explanatory factor is the underreporting of the number of portions or of the energy intake. Rippin et al. (2019) found that 32% and 44% of overweight adults were under-reporters of energy intake in the French INCA2 and UK NDNS studies, respectively [24]. The percentages of normal-weight under-reporters were much lower: 18% and 23%, respectively [24]. Other studies found few associations between food portion size and adiposity. The authors reported that the under-reporting of food intake could mask this association [20,21]. Moreover, Rippin et al. (2019) compared the consumption of energy-dense food by normal-weight and overweight subjects and observed that the consumption frequency of cake and chocolate was negatively associated with increasing BMI [24]. Because this result was unexpected, the authors supposed that there were high under-reporting levels, especially in the overweight and obese subjects. 

In the present work, the energy intake estimation is based on energy expenditure calculated by the eMouve algorithms published in [15] and [16] specifically developed for overweight and normal weight adults. The mean errors in absolute value for estimating energy expenditure are low: 5.7% and 8.5% in normal weight and overweight subgroups, respectively. The mean errors for estimating energy expenditure in relative value are null in both groups. That means that the estimations of energy intake at the group level (for both normal weight and overweight subjects) will be accurate. At the individual level and for normal weight subjects, the estimations for energy intake will also be accurate (plus or minus 5.7% of error). For an overweight subject, the estimation of energy intake is less accurate than for a normal weight subject, but with an error even lower than 10% (which is acceptable).

The first model of estimated EI (EI1) for all the subjects that included only the number of food portions gave poor results. The estimation of an energy intake by portion led to an error of 30%. This estimation led to an overestimation or an underestimation of total energy intake up to 1000 kcal/d compared to energy expenditure. This is not surprising considering the potential of both under-reporting and the large variation of portion sizes among the subjects. For this reason, we did not try to improve the accuracy of data collection because it is impossible to know the real values concerning the number and size of the portion. Thus, if a subject feels ashamed to report the consumption of energy-dense food, he/she forgets it consciously or unconsciously. We preferred to assess energy intake by a statistical data treatment, taking into account body weight, BMI status, and gender in food intake requirements and reports. 

By adding body weight in the regression models and separating subjects into four groups (normal-weight men and women, overweight men and women), we found that both body weight and the number of food portions played an essential role in the explanation of energy intake. The differences between estimated EI and TEE varied between 5 and 12% according to the subject group. In other studies, energy intake was underestimated up to 100% in a widely varied range [6,25]. Since the values of the regression coefficients and of body weight were high, body weight made a significant contribution in the evaluation of energy intake compared to the number of food portions, regardless of weight status and gender. In overweight subjects, the coefficients for the number of portions were higher than those in normal-weight subjects, meaning that a portion provided more energy in overweight than in normal-weight people. In other words, the size of the portion could be bigger or more energy-dense with increasing BMI. Regarding BMI, O’Brien et al. (2015) found varying results: in the NSIFCS 2001 study, the portion size of milk and butter was evaluated to be greater in obese subjects than in normal-weight subjects and vice versa in the NANS 2011 [21]. Pearcey and de Castro (2002) examined meal patterns and food intake of weight-stable and weight-gaining people and reported that the greater EI in the weight-gaining group was attributed to significantly larger meal consumption [26]. They also suggested that the dysregulation of food intake could be an integral component of weight gain. Similarly, Rolls et al. (2002) reported that the size of the food portion served at a lunch could significantly influence EI [27]. Burger et al. (2007) found that an individual’s BMI accounted for 28–51% of the variance in the choice of food portion size [28].

In the four regression models, one for each gender and BMI status, we improved not only the bias but also the agreement limits between the evaluation of EI and TEE. The bias was lower than 100 kcal, whereas it was close to −300 kcal in the first model. The best agreement limits of energy intake (460, 430 and 350 kcal/d) were observed for the subgroups of NW-M, NW-W, and OW-W of the test groups. The agreement limits for overweight men were larger, at 570 kcal/d. 

Very few studies have compared energy intake estimated by mobile applications with the TEE estimated by reference methods or research devices. In the work of Pendergast et al. (2017), the estimation of energy intake of a sample of young, normal-weight, and educated subjects was made by the FoodNow app on the basis of picture capture and a vocal message recording describing the food and beverages consumed [14]. The analyses of pictures and vocal messages were made in duplicate by trained nutritionists (double checking of each food code and amount). The energy intake estimated by this process was compared with TEE given by a very accurate research device (Armband). The bias was −197 kcal and the agreement limit between Armband TEE and EI was 425 kcal. In the present study, the bias is lower and the limits of agreement are similar or slightly higher. Our statistical approach to energy intake is competitive with those much more cumbersome and time-consuming analyses of food pictures. In contrast to many studies, we did not exclude the subjects who recorded a low number of portions (3 to 8 portions/d) or a high number of portions (20 to 30 portions/d). These small and big reporters represented 15.8% and 3.1% of the subjects, respectively. This way of reporting food consumption could be subject-dependent: it might be due to either infrequent reports with large-size portions or underreporting, or very frequent reports with small-size food portions. We adopted this point of view because we wanted our models to take account of both under- and overreporting, and for the various sizes of food portions. Obviously, our models are less suited to these small and big food reporters. 

The strength of this study is that it provides a high estimation of energy intake from four easily collectable variables: body weight, the total number of portions consumed by day, gender, and BMI group. Most of these variables are well known. The calculation is very quick and does not require nutritional knowledge. Moreover, this algorithm could be implemented in a new smartphone application dedicated to the general public. It could be useful for health professionals like dieticians to make a first estimation of a patient’s energy intake. It could also be useful to study large groups of subjects in epidemiological studies. This procedure is much simpler and cheaper, less time-consuming and intrusive for both patients/subjects and health professionals/researchers than the conventional methods.

The limitations of this study are the large agreement limits; however, they are no larger than those obtained by standard methods (food frequency, dietary report). Moreover, this algorithm is not food category-dependent and does not provide the energy intake estimation by food category. Furthermore, this study does not provide information about the food category that must be modified to improve weight status. If body weight does not change, the estimation of energy intake cannot vary to a large extent. It can only be used during a period of stable body weight and food consumption.

## 5. Conclusions

All four of the regression models based on body weight and the number of food portions were effective in estimating total energy intake in four subgroups of the population aged between 18–60 years. Knowing the individual body weight of the subjects made it possible to avoid the issue of food underreporting. Body weight is a major contributor in the four groups of subjects, and the number of portions provided a better explanation for more energy intake in the overweight subjects. These models may serve as an innovative and practical tool to estimate the total energy intake of the adult population.

## Figures and Tables

**Figure 1 foods-10-02273-f001:**
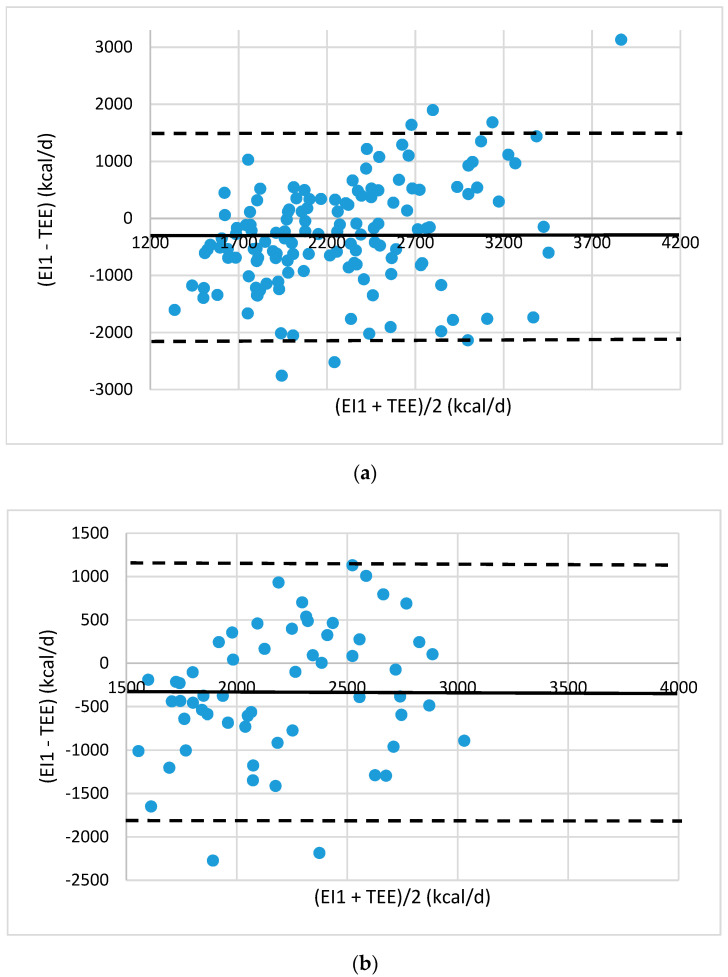
Bland–Altman plots of the agreement level between total energy expenditure (TEE) measured by eMouve, and energy intake (EI1) estimated by NutriQuantic in (**a**) the training and (**b**) the test group. Bias is represented as mean difference (two standard deviations).

**Figure 2 foods-10-02273-f002:**
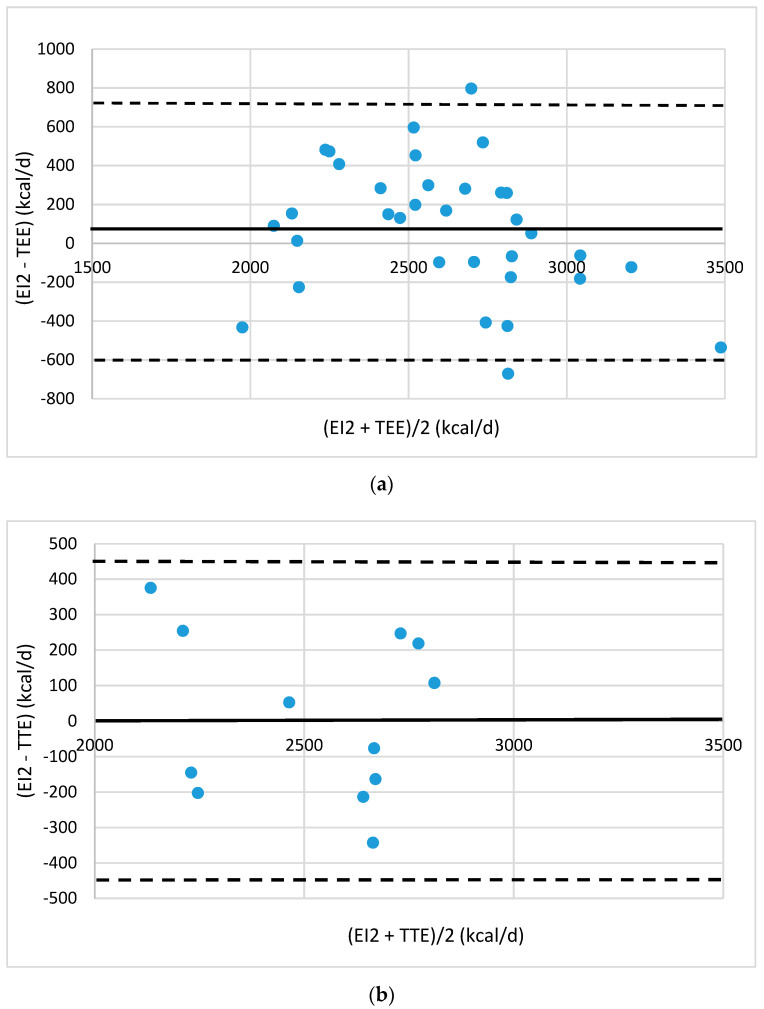
Bland–Altman plots of the agreement level between total energy expenditure (TEE) measured by eMouve, and energy intake (EI_2_) estimated by NutriQuantic in normal-weight men (**a**) in the training group and (**b**) in the test group. Bias is represented as mean difference (two standard deviations).

**Figure 3 foods-10-02273-f003:**
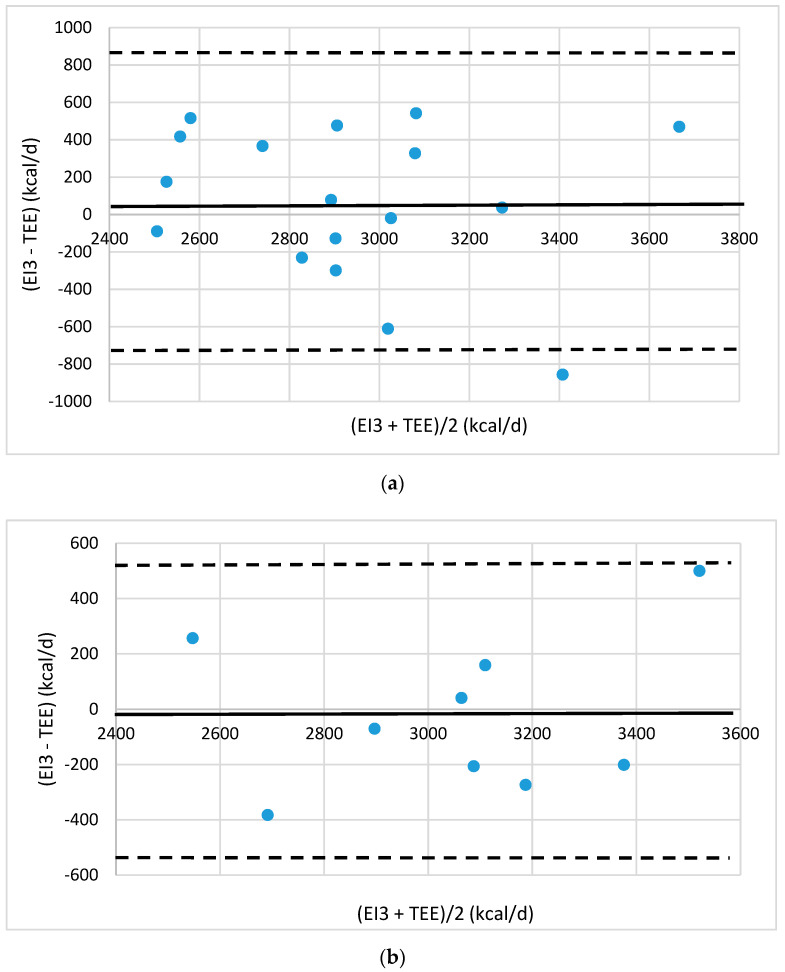
Bland–Altman plots of the agreement level between total energy expenditure (TEE) measured by eMouve, and energy intake (EI3) estimated by NutriQuantic in overweight-weight men (**a**) in the training group and (**b**) in the test group. Bias is represented as mean difference (two standard deviations).

**Figure 4 foods-10-02273-f004:**
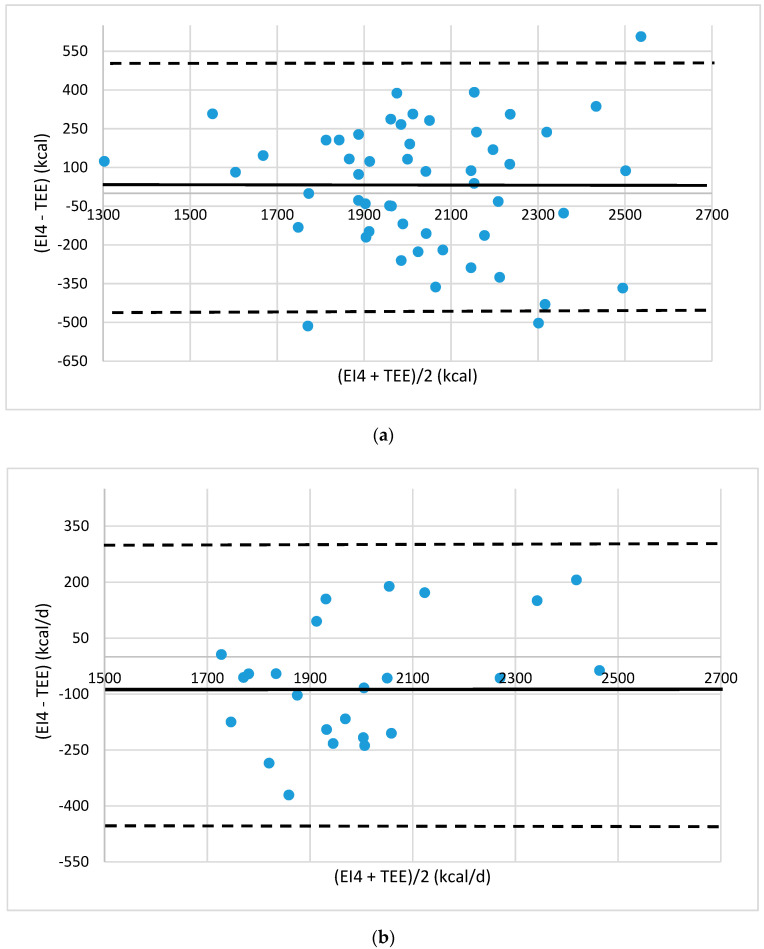
Bland–Altman plots of the agreement level between total energy expenditure (TEE) measured by eMouve, and energy intake (EI4) estimated by NutriQuantic in normal-weight women (**a**) in the train group and (**b**) in the test group. Bias is represented as mean difference (two standard deviations).

**Figure 5 foods-10-02273-f005:**
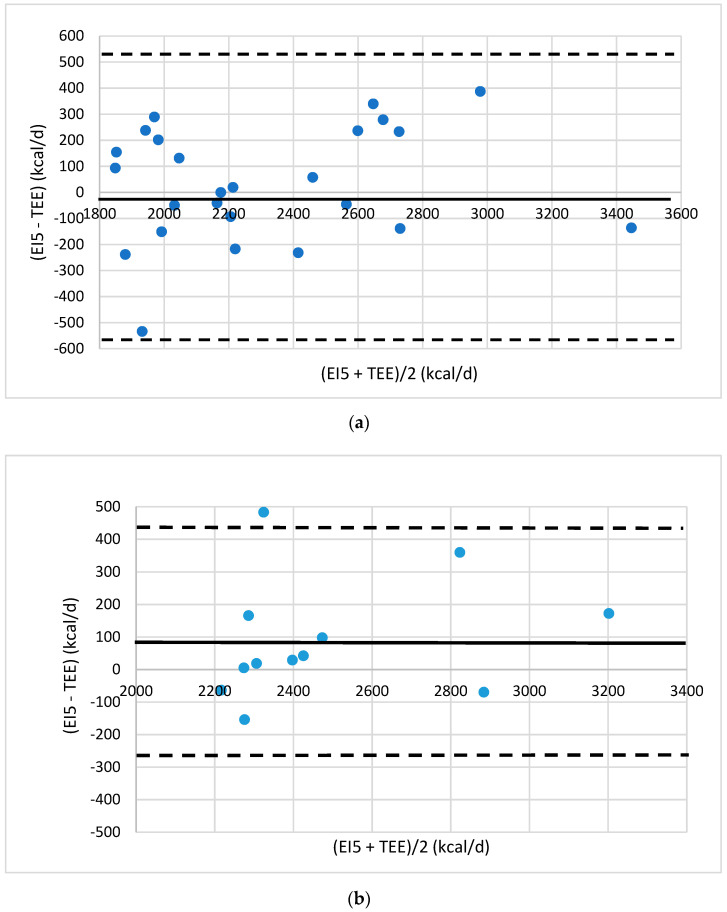
Bland–Altman plots of the agreement level between total energy expenditure (TEE) measured by eMouve, and energy intake (EI5) estimated by NutriQuantic in normal-weight women (**a**) in the training group and (**b**) in the test group. Bias is represented as mean difference (two standard deviations).

**Table 1 foods-10-02273-t001:** Characteristics of both training and test groups. (Mean values and standard deviations).

Group	Training	Test
Variable	Mean (SD)	Mean (SD)
Sample size (NW men)	35	12
Sample size (OW men)	18	9
Sample size (NW women)	52	25
Sample size (OW women)	25	13
Sex (% women)	60	64
Weight status (% NW)	65	63
Age (years)	37.5 (12.2)	37.4 (14.0)
Height (m)	169.8 (10.1)	168.8 (9.4)
Weight (kg)	71.1 (16.9)	71.6 (19.6)
BMI (kg/m^2^)	24.6 (5.7)	25.1 (6.6)

NW: normal weight, OW: overweight.

**Table 2 foods-10-02273-t002:** Effect of BMI status on behavioral data in men and women. (Mean values ± standard deviations, all the subjects).

Subgroup	NW-W	OW-W	NW-M	OW-M
Size	77	39	47	27
Variable	Mean (SD)	Mean (SD)	Mean (SD)	Mean (SD)
Number of portions/d	11.9 (3.9)	11.1 (4.1)	12.7 (5.3)	11.4 (3.9)
Nutritional balance score	6.1 (1.1)	5.6 (1.1) **	5.5 (1.4)	5.7 (1.2)
Total energy expenditure (kcal/d)	2080 (319)	2435 (505) ***	2610 (502)	2983 (486) ***

NW-W: normal-weight women, OW-W: overweight women, NW-M: normal-weight men, OW-M: overweight men; **, ***: Mean value was significantly different from that of the overweight group (*p* < 0.01, *p* < 0.001).

**Table 3 foods-10-02273-t003:** Mean energy contribution estimated from weight and the number of food portions observed in the training group. (Mean weight and number of portion and values of regression coefficients for body weight and portion according to gender and weight status).

Subgroup		Regression Coefficient Associated with		Energy Part (kcal) Explained by		Added Contribution (%) in OW Imputable to		Added Contribution (%) in OW Imputable to	
	Size	Weight (kg)	Portion (Nb)	Weight	Portion	Weight	Portion	Weight	Portion
NW-M	35	70.0	13.1	33.7	22.1	2359	289.5		
OW-M	18	93.1	11.2	27.1	36.7	2523	411.0	6.9	41.9
NW-W	52	58.1	11.7	30.3	22.3	1760	260.9		
OW-W	26	83.1	10.9	23.8	32.3	1978	352.0	12.4	34.9

NW-M: Normal-weight men, OW-M: Overweight men, NW-W: Normal-weight women, OW-W: Overweight women.

## Data Availability

The data presented in this study are available on request from the corresponding author.

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
