# Peer review of "Energy Intake Evaluation by a Learning Approach Using the Number of Food Portions and Body Weight"

_foods, 2021, doi:10.3390/foods10102273_

Round 1
Reviewer 1 Report
This study tried to develop a new estimation procedure of energy intake using simple variables. The idea is interesting and useful for many situations. However, the description of the method and results was confusing.
Major points
The authors made train and test groups. However, the use of these two groups was not clear. I guess that the train group was used to make the estimation model, and the test group was used to examine the accuracy of these estimation models. Was I right? However, if the models were made from the data of the train group, I think the estimation error would be more significant in the test group. But in this manuscript, the estimation error was greater in the train group. In addition, was it better to divide all subjects into a normal or obese groups than analyze all subjects together? I wonder the model estimating EI suddenly changed when the subjects become obese. Did the authors try to make equations to estimate EI using interaction of body weight (or BMI) and portion size?
Minor points
Introduction
Lines 74 to 77 Ref 14 was not a review.
Materials and methods
Lines 88 to 92 & table 1 Please indicate the number of the subject of normal weight, obese, and each sex for train and test groups.
Lines 101 to 102 The results of Ref 15 was 5.7(SD 4.6)% and Ref 16 was 8.5(SD 7.0), then “approximately 5%” seems too small.
Lines 103 to 105 Please explain more about the portion size with examples.
Lines 112 to 116 If the French legislation allowed to use of anonymous human data without ethical approval, please add this information here.
Lines 122 to 126 Did the authors compare 2 groups (normal weight and obese) or 4 groups (normal weight and obese in each sex)? If the authors compared between 2 groups, a one-way analysis of variance was not appropriate. If a one-way analysis of variance among 4 groups was used, which posthoc test was used?
Lines 152 to 166 I think this is a part of the instruction to the authors.
Results
Table 1, 2, 3 Please indicate the number of subjects in each group.
Table2 Is it a result of all subjects or a train group?
Table 3 I think this result is for the train group. If so, please include it in the title.
Discussion
Please add the discussion about accuracy or usefulness for individual and group data.
I think the estimation error of energy expenditure by eMouve can’t be over looked when it estimates individual energy expenditure. Please add the effect of the estimation error of eMouve in the discussion.
Reviewer 2 Report
- Big Error in manuscript text LINE 152-166 (Page 4), please remove. This might be copied from instruction for author.
- Table 1: there was “data” at the end of third column, typo should be removed.
- Table 2: The number in Table 2 should be divided in to male and gender (4 categories) instead of n=116, n=74 only
Round 2
Reviewer 1 Report
I think the revised version is adequately revised.
Author Response
The authors are happy that the revision is adequate.